

# A small venomous reptile from the Late Triassic (Norian) of the southwestern United States

Helen E. Burch[1], Hannah-Marie S. Eddins[1], Michelle R. Stocker[1], Ben T. Kligman[2], Adam D. Marsh[3], William G. Parker[3] and Sterling J. Nesbitt[1]

[1] Department of Geosciences, Virginia Polytechnic Institute and State University (Virginia Tech), Blacksburg, VA, United States
[2] Department of Paleobiology, National Museum of Natural History, Smithsonian Institution, Washington, D.C., United States
[3] Department of Science and Resource Management, Petrified Forest National Park, Petrified Forest, AZ, United States

Corresponding author
Helen E. Burch, heburch@vt.edu

## ABSTRACT

Reptile feeding strategies encompass a wide variety of diets and accompanying diversity in methods for subduing prey. One such strategy, the use of venom for prey capture, is found in living reptile clades like helodermatid (beaded) lizards and some groups of snakes, and venom secreting glands are also present in some monitor lizards and iguanians. The fossil record of some of these groups shows strong evidence for venom use, and this feeding strategy also has been hypothesized for a variety of extinct reptiles (*e.g.*, archosauromorphs, anguimorphs, and a sphenodontian). However, evidence of systems for venom delivery in extinct groups and its evolutionary origins has been scarce, especially when based on more than isolated teeth. Here, we describe a potentially venomous new reptile, *Microzemiotes sonselaensis* gen. et sp. nov., from a partial left dentary recovered from the Sonsela Member of the Chinle Formation (middle Norian, Upper Triassic) of northeastern Arizona, U.S.A. The three dentary teeth have apices that are distally reclined relative to their bases and the tip of the posteriormost tooth curves mesially. The teeth show subthecodont implantation and are interspaced by empty sockets that terminate above the Meckelian canal, which is dorsoventrally expanded posteriorly. Replacement tooth sockets are positioned distolingually to the active teeth as in varanid-like replacement. We identify this new specimen as a diapsid reptile based on its monocuspid teeth that lack carinae and serrations. A more exclusive phylogenetic position within Diapsida is not well supported and remains uncertain. Several features of this new taxon, such as the presence of an intramandibular septum, are shared with some anguimorph squamates; however, these likely evolved independently. The teeth of the new taxon are distinctively marked by external grooves that occur on the entire length of the crown on the labial and lingual sides, as seen in the teeth of living beaded lizards. If these grooves are functionally similar to those of beaded lizards, which use the grooves to deliver venom, this new taxon represents the oldest known reptile where venom-conducting teeth are preserved within a jaw. The teeth of the new species are anatomically distinct from and ~10x smaller than those of the only other known Late Triassic hypothesized venomous reptile, *Uatchitodon*, supporting venom use across multiple groups of different body

size classes. This new species represents the third Late Triassic reptile species to possibly have used envenomation as a feeding (and/or defensive) strategy, adding to the small number of venomous reptiles known from the Mesozoic Era.

# INTRODUCTION

Though the oldest diapsids known in the fossil record are as early as the Carboniferous Period (*Reisz & Müller, 2004*), it is not until the Triassic Period that these animals became pervasive and evolved a diversity of body sizes, *Bauplans*, and feeding ecologies (*e.g.*, *Brusatte et al., 2010*; *Turner & Nesbitt, 2013*; *Pritchard, 2015*; *Zanno et al., 2015*). New clades diversified as diapsid ecologies expanded, including archosauromorphs and lepidosauromorphs. Diverse feeding strategies evolved as diapsids radiated in the aftermath of the end-Permian mass extinction, including the use of venom to subdue prey.

A number of diapsids exhibit evidence of venom delivery, including two Triassic archosauromorphs (*Mitchell, Heckert & Sues, 2010*; *Sues, 1991*), a Jurassic sphenodontian (*Reynoso, 2005*), living beaded lizards and some varanoids and iguanians, ~2,500 species of living snakes, and close fossil relatives of living groups (*Fry et al., 2006*). Venom is used within these groups both for predation and defense, with venoms that primarily are used in defense having an increased ability to cause pain to deter predators, and venoms for predation that decrease motor function to prevent escape for prey capture and relocation (*i.e.*, having lethal neurological or coagulant effects; *Saviola, Peichoto & Mackessy, 2014*; *Koludarov et al., 2014*; *Schendel et al., 2019*). A variety of physical methods are utilized for venom delivery in Reptilia, including through saliva in lizards lacking grooved teeth (*e.g.*, anguid, varanid, and iguanian squamates; *Calvete et al., 2024*; *Fry et al., 2009a*, *2009b*), grooved teeth as seen in the archosauromorph *Uatchitodon kroehleri*, opisthoglyphous (*i.e.*, rear-fanged) colubrid snakes, *Sphenovipera jimmysjoyi*, and helodermatid lizards (*Reynoso, 2005*; *Koludarov et al., 2014*; *Mitchell, Heckert & Sues, 2010*; *Sues, 1991*), and injection *via* a tube within the tooth as seen in the archosauromorph *Uatchitodon schneideri* and solenoglyphous and proteroglyphous (*i.e.*, front-fanged) snakes like viperids (*Mitchell, Heckert & Sues, 2010*). Venom use is most common among snakes, and the hollow anterior fang mechanism for venom delivery has been largely conserved since its first appearance in the fossil record ~23 million years ago (*Kuch et al., 2006*). The presence of venom in extant lizards and snakes has given rise to the Toxicofera Hypothesis, which proposes that venom is ancestral to the clade Toxicofera that includes all squamates to the exclusion of lacertoids, scincoids, gekkotans, and dibamids (*Fry et al., 2006*, *2009a*; *Reeder et al., 2015*). However, this topic has been debated and is contradicted by anatomical data and homology in non-toxin molecular sequences (*Hargreaves, Tucker & Mulley, 2015*). Osteological correlates for venom in vertebrates typically include deeply grooved teeth for venom delivery, which are often the longest teeth in the jaw, sometimes with an apical opening connecting to a venom canal within the tooth, and a cavity or fossa (typically

within the maxilla) that may hold space for a venom duct, though this is not present in all venom-producing animals (*Benoit et al., 2017*; *Mitchell, Heckert & Sues, 2010*).

Within the fossil record, evidence of venom in early reptiles is exceedingly scarce and often is hypothesized only from isolated teeth such as in *Uatchitodon* (*Mitchell, Heckert & Sues, 2010*; *Sues, 1991*, *1996*). The oldest record of a structure for envenomation is seen in the Permian therapsid *Euchambersia mirabilis* (NHMUK R5696; *Benoit et al., 2017*), which possessed a strongly ridged incisiform dentition with deep grooves and a deep maxillary fossa to house a hypothesized venom gland. The dromaeosaur *Sinornithosaurus* has also been proposed as a venom-producing archosaur, the only suggested instance of venom production in archosauromorphs besides *Uatchitodon* (*Gong et al., 2010*). However, a reevaluation of specimens of *Sinornithosaurus* demonstrated its grooves teeth are dissimilar to grooves seen in living venomous species, and the proposed correlates for venom delivery are instead misinterpretations of anatomy and taphonomy (*Gianechini, Agnolín & Ezcurra, 2011*).

Here we describe unique venom-delivering teeth within a partial left dentary (DMNH PAL 2018-05-0017) representing a new genus and species of Late Triassic (Norian) reptile from the Sonsela Member of the Chinle Formation in northern Arizona, U.S.A. This specimen represents the earliest evidence of venomous teeth preserved within a jaw since the discovery of *Euchambersia* and demonstrates an uncommon feeding strategy in a Late Triassic community.

## MATERIALS AND METHODS

**Computed tomographic (CT) scanning parameters and segmentation**—We analyzed DMNH PAL 2018-05-0017 using X-ray microcomputed tomography (μCT-scanning) at the Shared Materials Instrumentation Facility at Duke University using a Nikon XTH 225 ST Scanner. The specimen was scanned at 185 kV and 76 μA with a 0.125 mm copper filter with 2,200 projections for each segment at a 14.30282 μm cubic voxel size. Reconstructions were created and analyzed using Mimics Innovation Suite 20. These data are available at Morphosource.org under Project 000607596. A 3D surface model was visualized from these data using MeshLab 2022.02. Teeth were imaged using a Hitachi TM3000 TableTop Scanning Electron Microscope with an accelerating voltage of 15 kV, a working distance of 5,800 μm, and 100x magnification. Permission for collection and study of this specimen was given by the Perot Museum of Nature and Science.

SYSTEMATIC PALEONTOLOGY
DIAPSIDA *Osborn, 1903* sensu *Gauthier, Kluge & Rowe, 1988*
*MICROZEMIOTES SONSELAENSIS*, gen. et sp. nov.
Figure 1

### Type Species—*Microzemiotes sonselaensis*

**Etymology**—The genus name *Microzemiotes* is derived from the Greek 'micro' = small, and 'zemiotes' = punisher. The species epithet *sonselaensis* recognizes the Sonsela Member of the Chinle Formation, which produced this specimen.

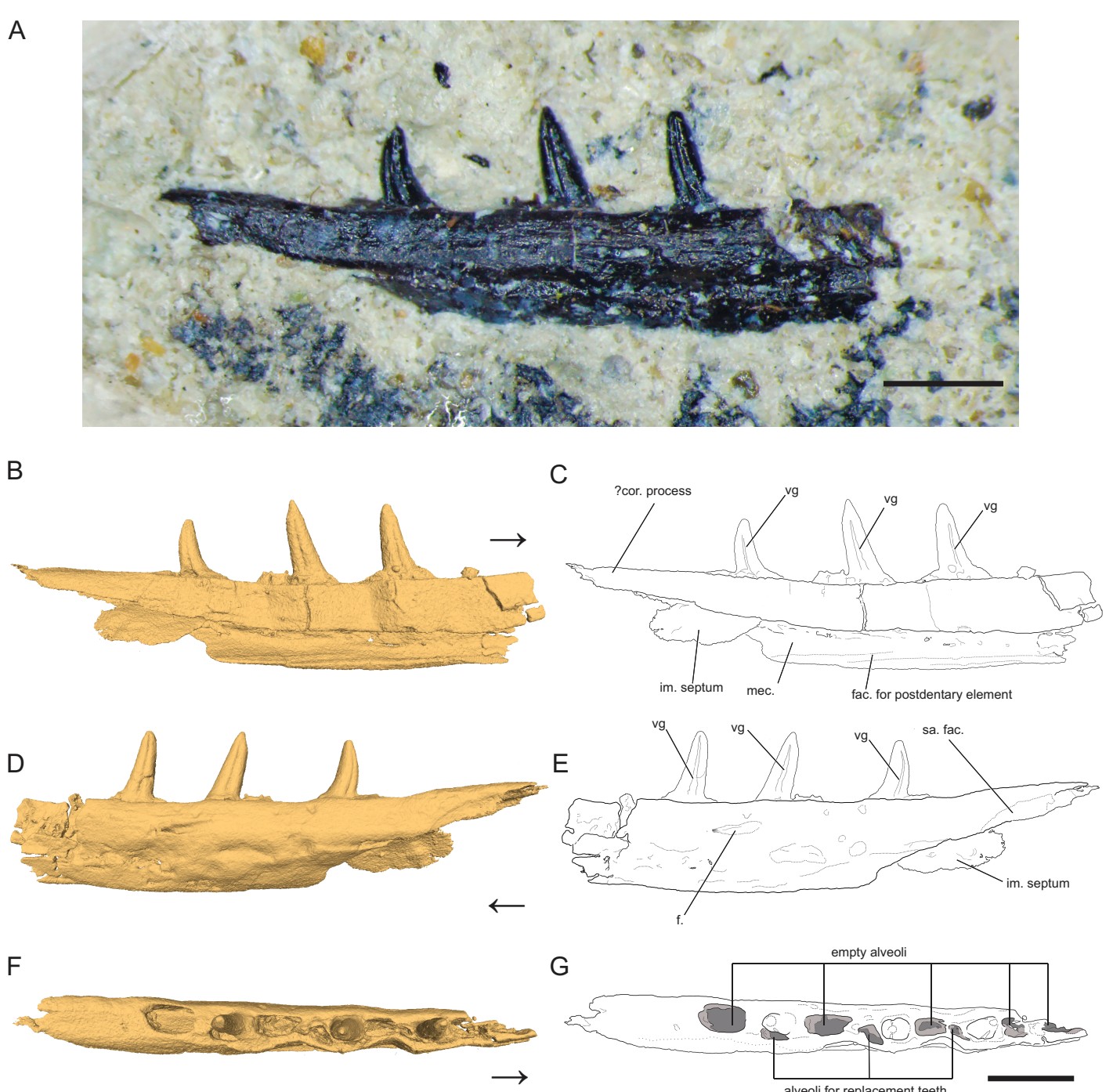

**Figure 1 Left Dentary (partial) of *Microzemiotes sonselaensis* holotype DMNH PAL 2018-05-0017.** (A) Photograph of *Microzemiotes sonselaensis* holotype DMNH PAL 2018-05-0017, (B, C) three-dimensional render and line drawing of the dentary in lingual view, (D, E) labial view, and (F, G) dorsal view. Scale bar equals 1 mm, arrow points anterior. Abbreviations: im, intramandibular; mec, Meckelian canal; fac, facet; f, foramen; sa, surangular; vg, venom groove.

**Holotype**—DMNH PAL 2018-05-0017, a partial left dentary with three preserved teeth, partially distorted on the lingual side.

**Diagnosis**—This species is diagnosed by the following combination of anatomical traits (potential autapomorphy denoted with an *): an ankylosed subthecodont dentition (*sensu Bertin et al., 2018*); distal dentary teeth are inclined distally from perpendicular to the mesial tooth edge; distal dentary teeth are oval in coronal cross-section (longer in the mesial-distal direction) with no carinae or serrations; replacement of teeth occurring in pits located distolingually to the fully-erupted teeth, a taller (1.0 mm) labial wall and slightly lower (0.9 mm) lingual wall of the dental shelf; intramandibular septum present; a rounded, incomplete, and ventrally-free intramandibular septum at the posteroventral portion of the dentary* (*sensu Estes, 1964*) projecting from the medial surface of the lateral wall; lingual and labial grooves extending from the base of the tooth to the tip of the crown.

**Locality, Horizon, and Age**—*M. sonselaensis* was recovered from the 'Green Layer' site, which is ~2–4 m of interbedded green and white laminated sandstone matrix (grain size ~0.5 mm) within the lower part of the Jim Camp Wash beds (*sensu Martz & Parker, 2010*) of the Sonsela Member of the Chinle Formation southeast of Petrified Forest National Park, Arizona (exact locality information on file at the Perot Museum of Nature and Science). The age of the locality is ~217.7 Ma–213.870 ± 0.078 Ma (*Kligman et al., 2020*; *Stocker et al., 2019*) based on local stratigraphic correlation with dated localities within Petrified Forest National Park, Arizona. Other vertebrate taxa present at the Green Layer include chondrichthyans (*e.g.*, *Reticulodus*), sarcopterygians (coelacanths and lungfish), and actinopterygians; tetrapods include salientians (*Stocker et al., 2019*), the allokotosaur *Trilophosaurus phasmalophos* (*e.g.*, DMNH PAL 2018-05-0012 and DMNH PAL 2018-05-0013; *Kligman et al., 2020*), leptosuchomorph phytosaurs, *Revueltosaurus callenderi* (*e.g.*, DMNH PAL 2018-05-0129), and aetosaurians (*e.g.*, DMNH PAL 2018-05-0014; *Mellett et al., 2023*). The presence of mystriosuchian leptosuchomorph phytosaurs and stratigraphic correlations with locality PFV 089 at Petrified Forest National Park suggest that the site occurs in the Revueltian estimated holochronozone (*Kligman et al., 2020*; *Martz & Parker, 2017*).

**Taphonomy**—The specimen was transported within a fluvial system and is preserved in a matrix-supported fine-grained sandstone with clasts of larger grains. Two of the *in-situ* teeth demonstrate an abrupt narrowing on the lingual side (0.2 mm basal from the tooth apex) inconsistent with our understanding of carnivorous teeth, which are generally conical, evenly tapering, and recurved (*Henderson, 1998*; *Jones, 2008*; *Presch, 1974*). Analysis using SEM shows this decrease in angle is the product of minute loss of fossil material (Figs. 2B and 2C). The enamel from the tooth apex could have been lost due to wear *via* occlusion and feeding use in life, or from fluvial transport processes prior to fossilization. The latter explanation is very unlikely because if degradation from fluvial transport occurred, it would be expected to cause damage to other parts of the jaw besides the tooth apices. We interpret the empty sockets as products of both biological and taphonomic processes; newly erupted replacement teeth in these alveoli were likely to have

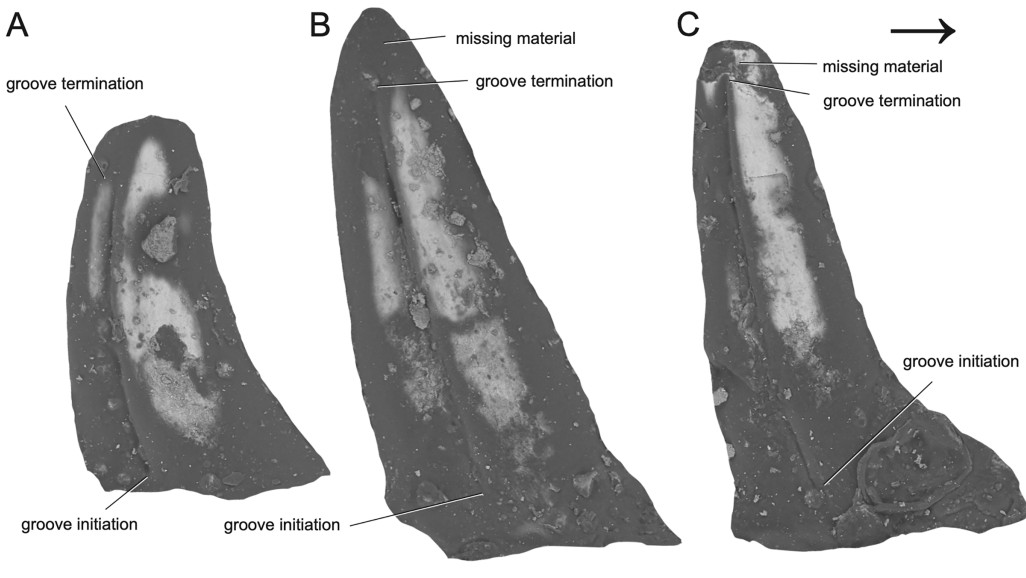

**Figure 2 SEM photographs of *Microzemiotes sonselaensis* holotype, DMNH PAL 2018-05-0017, grooved teeth (lingual view).** Teeth labelled (A–C) from distal to mesially. Scale bar equals 1 mm, arrow points mesially.

not been ankylosed and were therefore more susceptible to displacement, as in silesaurid dinosauriforms (*Mestriner et al., 2022*). The same is likely true for the replacement teeth developing in the dentary, which are marked by small cavities distolingual to the three parent teeth but lack *in-situ* replacements. The thin lingual wall of the dental shelf (0.05 mm in cross section) is deformed at each of its three contact points with the teeth, where compaction following burial caused lateral compression of the wall both into the empty sockets and across these hard *in-situ* teeth, resulting in breakage.

Nomenclatural acts—The electronic version of this article in Portable Document Format (PDF) will represent a published work according to the International Commission on Zoological Nomenclature (ICZN), and hence the new names contained in the electronic version are effectively published under that Code from the electronic edition alone. This published work and the nomenclatural acts it contains have been registered in ZooBank, the online registration system for the ICZN. The ZooBank LSIDs (Life Science Identifiers) can be resolved and the associated information viewed through any standard web browser by appending the LSID to the prefix http://zoobank.org/. The publication LSID is: urn:lsid: zoobank.org:pub:09D15F7E-D5AD-4AC0-BE94-8B7FB5A6DDCF. The online version of this work is archived and available from the following digital repositories: PeerJ, PubMed Central SCIE and CLOCKSS.

# DESCRIPTION

## Dentary

The holotype of *Microzemiotes sonselaensis* (DMNH PAL 2018-05-0017) consists of a partial left dentary with three well preserved teeth (Fig. 1) that is distorted by lateral

post-depositional compression on the medial surface. The dentary is broken and missing bone anterior to the first tooth position (counting alveoli from anterior to posterior). The preserved portion is 6 mm long anteroposteriorly and 1 mm deep dorsoventrally (measured from CT data in Mimics v.20), with teeth rising ~0.8 mm above the labial wall of the dentary shelf, which is perforated with a foramen. The posterior process of the dentary extends 1.33 mm beyond the distal edge of the terminal tooth socket and narrows dorsoventrally to a point. The termination is dorsally inflected 4° above the dorsal surface of the tooth-bearing portion of the dentary, creating a concave effect. The overall anatomy of the dentary is similar to those of early diverging diapsids: the dorsally inflected posterior taper of the dentary is also seen in *Youngina capensis* (BP/1/2871) but unlike that specimen it does not bifurcate into posterolateral and posteromedial processes (*Hunt et al., 2023*). A posteriorly tapering dentary is also present in tanystropheids including *Tanystropheus hydroides* (PIMUZ T 2790), *Tanystropheus longobardicus* (PIMUZ T 3901), and *Macronemus bassanii* (PIMUZ T 2477), the archosauromorph *Prolacerta broomi* (UCMZ 2003.41R), Permian varanopids such as *Mesenosaurus efremovi* (ROMVP 85456), *Varanops brevirostris* (FMNH UC 64), and *Aerosaurus wellesi* (UCMP 40096) and parareptiles such as *Feeserpeton oklahomensis* (OMNH 73541), *Colobomycter pholeter* (BMNRP 2008.3.1), and *Carbonodraco lundi* (CM 41714) (*Campione & Reisz, 2010*; *Hunt et al., 2023*; *Macdougall et al., 2017*; *MacDougall et al., 2019*; *Maho, Gee & Reisz, 2019*; *Mann et al., 2019*; *Miedema et al., 2020*; *Nosotti, 2007*; *Sobral, 2023*; *Spiekman et al., 2020*; *Langston & Reisz, 1981*). A tapered dorsal process of the posterior portion of the dentary that is accompanied by ventral projections of the dentary (*e.g.*, subdental shelf, angular process, surangular process, posteroventral process) that extend posteriorly to or nearly to the end of the coronoid process forming a V or W shape occurs in some archosauromorph taxa as well as some squamate groups, most notably the scincoids (*Ezcurra, 2016*; *Hutchinson, Skinner & Lee, 2012*; *Spiekman, Fraser & Scheyer, 2021*). Ventral processes are absent in the posterior portion of the dentary of *Microzemiotes sonselaensis*, which instead possesses a blunt end to the medially curved subdentary shelf. No sockets are present on this posterior portion in *Microzemiotes sonselaensis*, and a shallow, curved facet is present on the ventral edge of the lateral surface, most likely articulating with the surangular as seen in the diapsid *Youngina capensis* (BP/1/2871) and the archosauromorphs *Prolacerta broomi* (UCMZ 2003.41R) and *Macronemus bassanii* (PIMUZ T 2477) which seem to have homologous arrangements of mandibular anatomy (*Hunt et al., 2023*; *Miedema et al., 2020*; *Sobral, 2023*). The lateral surface of the dentary of *Microzemiotes sonselaensis* is convex, as the ventral half curves medially beneath the tooth bearing dorsal surface. The ventral edge of the lateral wall of the dentary of *Microzemiotes sonselaensis* is medially inflected to form the floor of the Meckelian canal. There is a concave ridge on the medial surface of this ventral edge of the dentary that we interpret as an articular surface for the splenial as in *Youngina capensis* (BP/1/2871; *Hunt et al., 2023*). The Meckelian canal is prominent and is medially open for the preserved length (presumed to be enclosed by the splenial in life) and widens dorsoventrally in the posterior direction, as is the condition in many amniotes. The anatomy of the anterior portion of the dentary and symphysis is unknown.

The Meckelian canal is incompletely divided by an intramandibular septum, which forms a round C-shaped edge projecting posteriorly from between the lateral and medial walls of the dentary (Figs. 1B and 1C). This edge connects to the medial surface of the lateral wall of the dentary ventral to the two posteriormost teeth, creating brief separation of the alveolar cavity from the Meckelian canal. In *Microzemiotes sonselaenisis*, the intramandibular septum projects from the medial surface of the lateral wall (Fig. 3A3) rather than extending from the ventral surface of the medial wall to lower on the medial surface of the lateral wall, as is the condition in anguid lizards and snakes (*Lee & Scanlon, 2001*). Among Triassic reptiles, a similar posteriorly projecting intramandibular septum is only known from the kuehneosaurid-like late Carnian diapsid *Idiosaura virginiensis* (*Kligman, Sues & Melstrom, 2024*); however, differences in the septums' divisions of the inferior alveolar canal and Meckelian canal (divided for *Idiosaura virginiensis*, undivided for *Microzemiotes sonselaensis*), in addition to differences in tooth attachment and neurovascular morphology, suggest that this similarity is likely convergent, and is otherwise unknown among Triassic archosauromorphs.

## Dentition

The dentary preserves eight tooth positions including three pits for replacement teeth located distolingually to the fully erupted teeth. The three teeth present, in sockets three, five, and seven identified from the anterior (Figs. 1F, 1G, 3) are conical in sagittal cross section and oval (mesiodistally longest) in coronal cross section. The distal dentary teeth do not erupt perpendicular to the dentary but are inclined distally 20–24° from perpendicular toward the mesial tooth edge. The third tooth (distalmost) is recurved mesially. Though mesially curving teeth are scarcely documented, they are not entirely unheard of in the distal teeth of some reptilian groups that show more typical distally recurved teeth or straight teeth in the rest of the jaw (*e.g.*, the dinosaur *Camarasaurus*, SMA 0002 and the diapsid *Maiothisavros dianeae*, ROMVP 87366 (*Mooney et al., 2022*; *Wiersma & Sander, 2017*)). Thus, this single tooth may not indicate mesial curvature for the rest of the more mesial teeth. The teeth of *Microzemiotes sonselaensis* lack carinae or serrations but possess deep labial and lingual grooves that span from just dorsal to the connection to the dentary up to the apex of the crown on the labial side and up to 0.1–0.2 mm away from the tooth apex where original fossil material is missing on the lingual side (Fig. 2).

The teeth in *Microzemiotes sonselaensis* exhibit subthecodont implantation, inset in sockets extending 60% of the dorsoventral depth of the dentary measured from the lateral side, underlain by the Meckelian canal (Figs. 1B, 1C, 3A). The lateral wall is slightly higher than the medial wall, as seen in *Youngina capensis*, which also exhibits subthecodont dentition (*Hunt et al., 2023*). The sockets lacking *in-situ* teeth are incomplete ventrally and open directly into the Meckelian canal, which is uncommonly described in taxa with subthecodont implantation but has been noted in mosasaurs such as *Clidastes propython* (FMNH PR 164, PR 4; *Rieppel & Kearney, 2005*), but this may be a product of taphonomy. The roots of the teeth of DMNH PAL 2018-05-0017 are 0.4–0.5 mm long measured apicobasally, covered medially by the medial wall. The tooth roots are completely

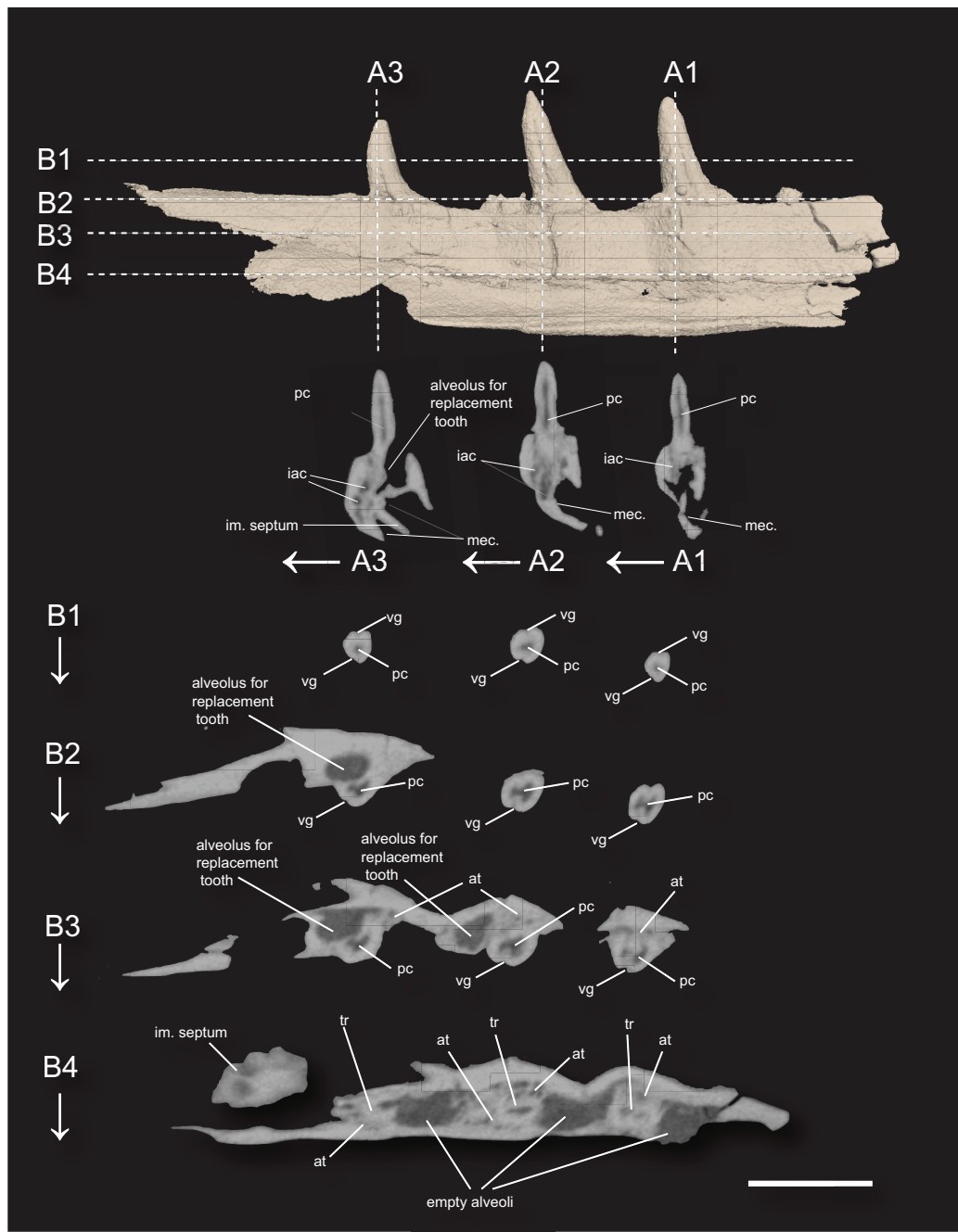

**Figure 3** **3D surface rendering of the dentary of *Microzemiotes sonselaensis* holotype DMNH PAL 2018-05-0017 in lingual view and CT cross sections.** (A1–A3) Series of coronal sections and (B1–B4) series of axial sections of the CT reconstructions (slices) showing internal anatomy. Abbreviations: at, ankylosing tissue; iac, inferior alveolar canal; im, intramandibular; mec, Meckelian canal; tr, tooth root; pc, pulp cavity vg, venom groove. Scale bar equals 1 mm, arrows indicate labial direction.

surrounded by ankylosing tissue (Figs. 3B3, 3B4) as in ankylothecodont archosauromorphs like allokotosaurs and silesaurids (*e.g.*, *Mestriner et al., 2022*).

The dentary of *Microzemiotes sonselaensis* shows an alternate method of tooth replacement, with the three *in-situ* teeth interspaced with empty sockets, and alveoli for replacement teeth are positioned distolingually to active teeth (*Bertin et al., 2018*; Fig. 3B). The alternate replacement method is noted in a variety of groups including early diverging diapsids like *Youngina capensis*, archosauromorphs such as *Prolacerta broomi*, Triassic sauropterygians like *Nothosaurus* and *Simosaurus*, and in many modern lizard groups (*Gow, 1974*; *Neenan et al., 2014*; *Shang, 2007*). Though the alternate method of replacement is clear in this portion of the dentary, replacement strategies may be variable throughout the jaw, as has been noted in *Youngina capensis* and *Prolacerta broomi* (*Hunt et al., 2023*; *Sobral, 2023*).

## DISCUSSION

### Proposed taxonomy

DMNH PAL 2018-05-0017, the holotype and only known specimen of *Microzemiotes sonselaensis*, possesses few clear character states that indicate a phylogenetic affinity within Amniota. We attribute this new taxon to the sauropsid group Diapsida on the basis of a combination of character states highlighted most recently by *Ford & Benson*'s *(2019)* character list used in their 2020 analysis of amniote phylogeny: teeth, marginal dentition, cutting edges (carinae) on the mesial and distal surfaces: absent ((character) 8– (state) 0); teeth, serrations on crown of marginal teeth: absent (9–0); teeth, multiple apical cusps on marginal dentition: absent (11–0). Though these states may be plesiomorphic for early-diverging diapsids and synapsids alike, when taken in the context of the temporal and geographic setting for this specimen, we find it most likely that *Microzemiotes sonselaensis* is a diapsid reptile. The posteriorly tapering and dorsally inflected shape of the dentary seen in *Microzemiotes sonselaensis* is also present in early-diverging diapsids from the Permian (*Youngina*, varanopids, and parareptiles), though no varanopids are known to have persisted into the Triassic Period and the only Triassic parareptiles are the procolophonids, which have not yet been documented from the Sonsela Member of the Chinle Formation, and those found elsewhere in the Late Triassic of North America (leptopleuronine procolophonids) bear a suite of strikingly different morphologies (*e.g.*, *Mueller et al., 2024*; *MacDougall, Brocklehurst & Fröbisch, 2019*). It should be noted that varanopids were previously identified as synapsids; however, a recent phylogenetic hypothesis using extensive morphological characters recovered varanopids as early diverging diapsids instead (*Ford & Benson, 2020*). Subthecodont dentition is shared among these taxa and *Microzemiotes sonselaensis*, a condition also seen in mosasaurs, sauropterygians, icthyopterygians, and non-therapsid synapsids, and has been suggested as the plesiomorphic condition for both Diapsida and Amniota (*Bertin et al., 2018*; *Caldwell, 2007*; *Evans, Maddin & Reisz, 2009*; *Liu et al., 2016*; *Sues & Kligman, 2020*).

Within Diapsida, the phylogenetic placement of *Microzemiotes sonselaensis* is ambiguous given the limited number of observable character states in the only known specimen that are apomorphic of various clades. However, *Microzemiotes sonselaensis* does

share anatomy that among living groups is only found in some lizards. The presence of an intramandibular septum with a free posteroventral margin, as noted in *Microzemiotes sonselaensis*, has been described as an unambiguous synapomorphy for extant anguids within Squamata, and besides *Idiosaura virginiensis*, all archosauromorphs and non-squamate lepidosauromorphs lack this feature (*Estes, Pregill & Camp, 1988*; *Conrad et al., 2011*; *Kligman, Sues & Melstrom, 2024*). The extension of the intramandibular septum posterior to the distalmost teeth would characterize the septum as being 'well developed' (*Estes, Pregill & Camp, 1988*), a character state that is considered a synapomorphy of anguimorphs and convergent in some iguanians (*Lee & Scanlon, 2001*). However, the intramandibular septum in *Microzemiotes sonselaensis* differs from those seen in most anguimorphs in two key ways: 1) the ventral margin is not coossified to the body of the dentary; and 2) the anterior edge terminates under the second tooth from the posterior end, poorly separating the alveolar foramen from the Meckelian canal. These differences demonstrate that convergence of this structure is more likely than homology with anguimorphs. It is notable that the posterior projection of the intramandibular septum forms a C shape, which although independently evolved, is also seen in the intramandibular septum of some anguid lizards such as the Pleistocene *Ragesaurus medasensis* (USTL MED-121; *Bailon & Auge, 2012*) and those of agamid lizards, particularly *Uromastyx aegyptia* (SAMA R48106; *Hutchinson, Skinner & Lee, 2012*). The intramandibular septum also has a somewhat enigmatic nature as a character due to challenges preparing specimens to reveal the fragile septum leading to inconsistent descriptions of the character relative to variable jaw anatomy. The intramandibular septum of the Triassic diapsid *Idiosaura virginiensis* suggests that this anatomy likely evolved multiple times in Triassic diapsids (*Kligman, Sues & Melstrom, 2024*).

*Microzemiotes sonselaensis* shares several distinctive characters with the tooth morphotype named *Uatchitodon*, including recurved, labiolingually compressed teeth, and venom grooves (specifically *Uatchitodon kroehleri*, USNM542518; *Mitchell, Heckert & Sues, 2010*); however the teeth of *Microzemiotes sonselaensis* are much smaller than those of any specimen of *Uatchitodon* and lack serrations. Because reptile tooth implantation and anatomy are known to change with shifts in diet through ontogeny (including the presence/absence of serrations), we must consider the possibility that *Microzemiotes sonselaensis* may be an early ontogenetic form of *Uatchitodon schniederi*, the spatiotemporally close *species* of *Uatchitodon* (*Cipriani et al., 2017*; *Codron et al., 2012*; *Farlow et al., 1991*; *Griffin et al., 2021*; *Maho & Reisz, 2024*; *Mitchell, Heckert & Sues, 2010*; *Tucker et al., 1996*). To assess this possibility, we compare to the development of venom conducting teeth in living viperids and elapid snakes that possess venom conducting tubes like *Uatchitodon schniederi*. The development of fangs is expressed apically with an opening already present, and tooth growth is basal, rather than a grooved fang becoming infolded to create a tube, meaning all ankylosed teeth already possess a complete tube (*Jackson, 2002*; *Vonk et al., 2008*). If the fangs of *Uatchitodon schniederi* followed a similar developmental pathway, a grooves-only stage for the teeth would not exist, even in ontogenetically young forms. Based on this rational, we assert that DMNH PAL 2018-05-0017 is likely a separate species from *Uatchitodon schniederi*.

## Evidence of venom in *Microzemiotes sonselaensis*

The teeth of *Microzemiotes sonselaensis* possess a suite of osteological correlates that indicate that it may have utilized envenomation as a feeding or defensive strategy, and what follows is a discussion on the evidence to support this hypothesis. The grooves on the lateral surfaces of all three teeth present in the dentary of *Microzemiotes sonselaensis* extend from contact with the dorsal surface of the dentary to the tips of the teeth and are only absent where surface enamel was lost due to breakage (Fig. 2C). We rule out the possibility that these grooves are the product of wear based on inspection of SEM photos (Fig. 2) that show the tooth enamel to be fully intact across the grooves and that the shape of the grooves remains constant across the lingual and labial sides and across the three preserved teeth. The presence of deep grooves extending from the base to the tip of the tooth in a non-mammalian amniote is strongly indicative of the presence of an envenomation system (*Mitchell, Heckert & Sues, 2010*). The continuation of the groove from the base to the tip of the tooth is consistent with the venom delivery structure present in helodermatid lizards, which use a combination of capillary action and a sharp cutting surface to deliver (inject implies a pressurized system) venom into prey through a sustained bite (*Koludarov et al., 2014*). The teeth of *Microzemiotes sonselaensis* are similar to the conical, curved, and deeply grooved teeth of the extant venom-producing Gila monster (*Heloderma suspectum*), which are characterized by deep surficial venom grooves and lack interior venom canals with apical openings, though the teeth of the Gila monster have a single groove (located on the mesial surface) on each tooth in the maxilla and dentary rather than grooves on both the labial and lingual surfaces. Similarly, deep labial grooves for venom conduction are present in the maxillary teeth of opistoglyphous (*i.e.*, rear-fanged) snakes; grooved teeth primarily occur in the posterior end of the maxilla and are variable in number and shape across species and ecology (*Cleuren, Hocking & Evans, 2021*; *Westeen et al., 2020*). The venom conducting teeth of *M. sonselaensis, H. suspectum*, and opistoglyphous snakes have a less complex condition for venom delivery, which is also observed in helodermatids, *Uatchitodon kroehleri*, and solenoglyphous and proteroglyphous snakes' early stage fangs that lack internal tubes (*Mitchell, Heckert & Sues, 2010*). The location of the venom gland varies among these animals. In helodermatids, the venom gland is located anterolaterally to the dentary and venom is secreted to the bases of the grooved dentary teeth through ducts; the maxillary teeth are also grooved but do not have connected ducts (*Fry et al., 2006*; *Mackessy, 2022*). In venomous snakes (colubrids, elapids, and viperids) the gland, or group of glands, is located subdermally ventral to the eye (*Mackessy, 2022*; *Schendel et al., 2019*). The location of the venom gland is unknown for *Uatchitodon kroehleri* and *Uatchitodon schneideri*, which are only represented by teeth (*Mitchell, Heckert & Sues, 2010*). We suggest the position for a venom gland for *Microzemiotes sonselaensis* would be anterolateral to the dentary with ducts leading to the bases of the grooved teeth, as is the condition in helodermatids which also possess many grooved teeth within the mandibles to supply venom slowly for a sustained bite.

Regardless of delivery method, venom can function to disable prey or to defend against attacking predators. In helodermatids, venom is hypothesized to function both defensively

and for predation, whereas in viperous snakes it is used primarily for predation (*Saviola, Peichoto & Mackessy, 2014*; *Koludarov et al., 2014*; *Schendel et al., 2019*). If the venom delivery system in *Microzemiotes sonselaensis* was functionally similar to that of helodermatids and opitoglyphous snakes as suggested by shared anatomy, it would follow that venom delivery was more passive compared to taxa that inject venom *via* interior tubes (*e.g.*, vipers and possibly *Uatchitodon schneideri*), and an individual would need to grasp its target for some amount of time (*e.g.*, up to 1 h in helodermatids) in order to inflict significant damage (*Koludarov et al., 2014*). However, though the method of delivery may be inferred from fossil evidence, the active chemical components of the venom itself cannot be assessed without soft tissue and/or biomolecules and so any biochemical mechanism of venom in *Microzemiotes sonselaensis* remains unknown.

## Implications for venomous reptiles in the Late Triassic

Prior to the discovery of DMNH PAL 2018-05-0017, anatomy consistent with a venom apparatus was observed in only two other Triassic taxa, *Uatchitodon kroehleri* and *Uatchitodon schneideri*, both from Late Triassic (Carnian and Norian, respectively) deposits of North America. Occurrences of *Uatchitodon schneideri* in western North America include the following: UCMP A269/MNA loc. 207 (the *Placerias* Quarry; *Mitchell, Heckert & Sues, 2010*; *Sues, 1996*); PFV 396 (the Coprolite Layer; *Parker et al., 2021*); and PFV 456 (Thunderstorm Ridge; *Kligman, 2023*). The *Placerias* Quarry has a maximum depositional age of 219.39 ± 0.16 Ma (*Ramezani, Fastovsky & Bowring, 2014*), whereas the Green Layer has an estimated age of ~217.7 Ma to 213.870 ± 0.078 Ma (*Kligman et al., 2020*; *Stocker et al., 2019*) (Fig. 4). The ages of these localities and assemblages present indicate that the two species are separated by the Adamanian-Revueltian boundary, a time of potentially significant faunal turnover in North America (*Parker & Martz, 2011*). *Uatchitodon schneideri* is restricted to the Adamanian teilzone whereas *Microzemiotes sonselaensis* is present in the Revueltian teilzone (*Martz & Parker, 2017*). However, with minimal distance and time separating these specimens, the co-occurrence of *Uatchitodon schneideri* and *Microzemiotes sonselaensis* cannot be ruled out. Both species of *Uatchitodon* are described only from isolated teeth and have been thought to represent carnivorous archosauromorphs based on the presence of compound denticles (*i.e.*, serrations, or denticles with divided or irregular cutting edges) and thecodont implantation based on a single specimen possessing a root, USNM 448624 (*Sues, 1991*, *1996*). Serrated teeth have evolved many times throughout the fossil record but are only recognized in one group of venom-producing reptiles, the varanids, including *Varanus komodoensis* (lacking labial and lingual grooves) and the extinct *Varanus* ('*Megalania*') *priscus* (possessing labial and lingual grooves) that both possess ziphodont teeth with structures for venom delivery (*Fry et al., 2009b*). Functionally, ziphodont teeth allow for slicing of prey tissue, suggesting this was part of the feeding ecology of *Uatchitodon*. The absence of ziphodonty in *Microzemiotes sonselaensis* suggests that unlike *Uatchitodon*, it may have used its teeth for piercing prey tissue only, not slicing. *Microzemiotes sonselaensis* and both species of *Uatchitodon* are characterized by conical, sharp teeth with both labial and lingual grooves, a character state that has not

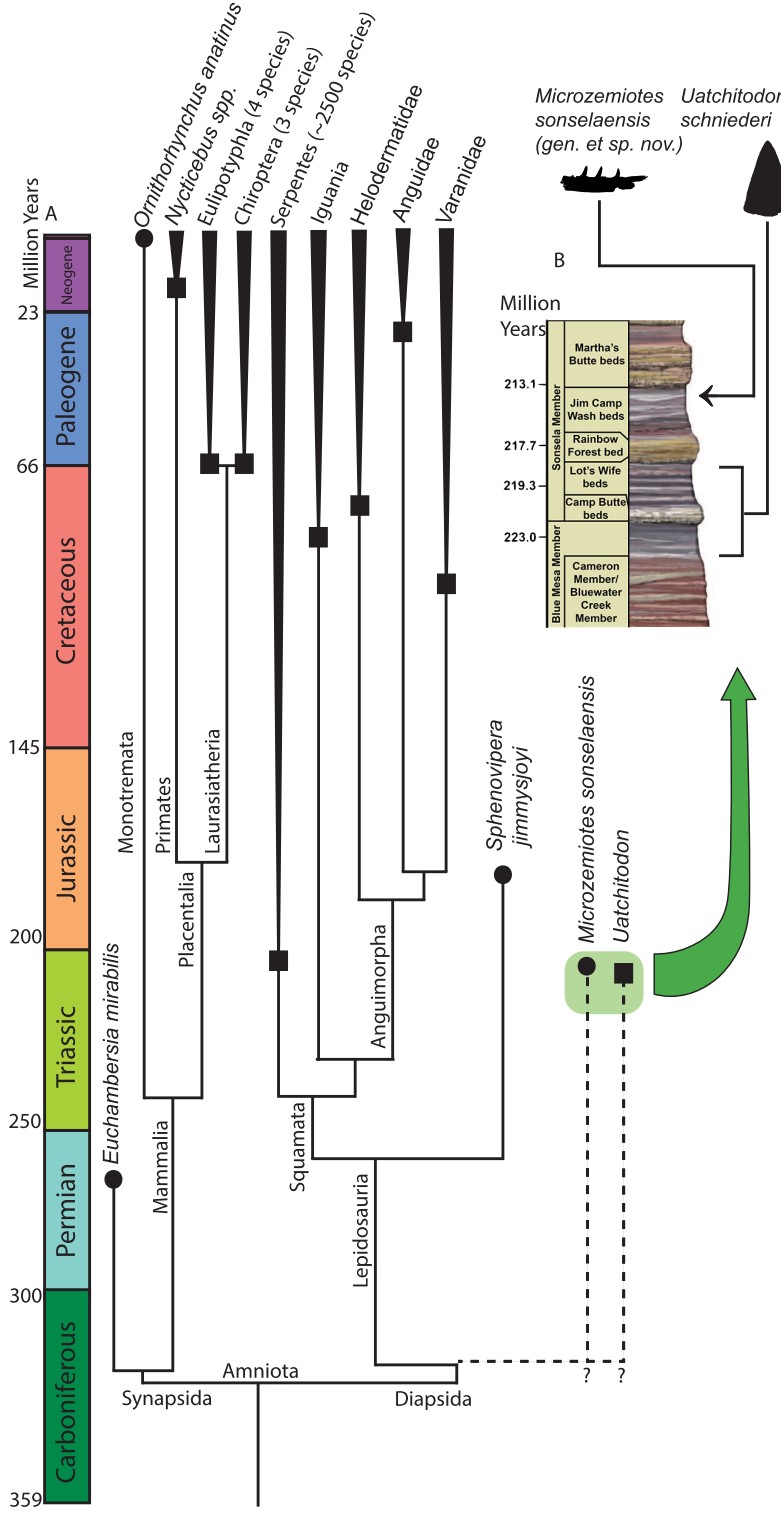

**Figure 4 Phylogenetic tree of venom producing taxa and stratigraphic context.** (A) Phylogenetic tree of venom producing vertebrate taxa modified from *Ford & Benson (2020)*, *Rougier, Martinelli & Forasiepi (2021)*, *Upham, Esselstyn & Jetz (2019)*. Black circles indicate first known occurrence of a venom producing taxon, black squares indicate first appearance of venom producing group. (B) Stratigraphic context of occurrences of *Microzemiotes sonselaensis* in the Green Layer Locality and *Uatchitodon schneideri* in the

**Figure 4** (continued)
*Placerias* Quarry (*Sues, 1996*; *Mitchell, Heckert & Sues, 2010*); PFV 396 (Coprolite Layer; *Parker et al., 2021*) ; and PFV 456 (*Kligman, 2023*) in the Chinle Formation in Arizona, USA. Outlines of *M. sonselaensis* and *U. schneideri* (modified from *Sues (1996)*: https://doi.org/10.1080/02724634.1996.10011340) are scaled to one another. Stratigraphic column modified from the National Park Service (https://www.nps.gov/pefo/learn/nature/geologicformations.htm; Public Domain).

been observed in living reptilian venom users. Coronal cross sections of the teeth (Figs. 3B2, 3B3) of *Microzemiotes sonselaensis* are similar to those of *Uatchitodon kroehleri* and *Uatchitodon schneideri* in that the pulp cavity is compressed at the center of the tooth in the labio-lingual direction (*Mitchell, Heckert & Sues, 2010*). In the teeth of *Uatchitodon schniederi* and some specimens of *Uatchitodon kroehleri*, the pulp cavity is so compressed that it is divided, which is not seen in any of the teeth of *Microzemiotes sonselaensis* (*Mitchell, Heckert & Sues, 2010*). The addition of *Microzemiotes sonselaensis* to the recognized venom producing taxa of the Late Triassic indicates that venom was likely utilized by taxa with some variations in tooth morphology.

One specimen of *Uatchitodon schneideri* (MNA V3680; *Mitchell, Heckert & Sues, 2010*; *Sues, 1996*) recovered in northeastern Arizona consists of an isolated tooth that measures ~6.5 mm from base to tip, longer than the entire preserved length of the dentary in *Microzemiotes sonselaensis*, which holds teeth measuring no more than 0.8 mm from base to tip (size comparison in Fig. 4). Some of the smallest extant vertebrate venom producers are among the opisthoglyphous members of the colubrid family, which use venom in a similar method to helodermatids with grooved fangs in the posterior region of the maxilla and venomous saliva to deliver a sustained bite (*Fry et al., 2009a*). Snakes benefit from venom use in the ability to subdue large prey that they can consume using flexible hemimandibles and a ligamentous mandibular symphysis. Small mammalian venom users, such as the short tailed shrew (7–10 cm long), lack this ability and have been noted to use venom for the immobilization of prey (insects and mice) for ease of consumption or for delayed feeding (*Martin, 1981*; *Schendel et al., 2019*; *Tomasi, 1978*). The diminutive size of the preserved length of the dentary in *Microzemiotes sonselaensis* suggests that this animal was a very small predator and likely had even smaller prey – potentially insects and similarly small vertebrates. Wear from feeding during life at the tooth apices suggests that it may have fed on invertebrates with exoskeletons (it clearly wasn't eating mussels). We demonstrate here that *Microzemiotes sonselaensis* was a much smaller predator than *Uatchitodon*, supporting venom use among different size classes of predators in the Late Triassic.

## INSTITUTIONAL ABBREVIATIONS

**BMRP**    Burpee Museum of Natural History, Rockford, Illinois, U.S.A.

**BP**    Bernard Price Evolutionary Studies Institute, University of Witwatersrand, Johannesburg, South Africa

**BRSMG**    The Bristol City Museum, Bristol, U.K.

**CM**    Carnegie Museum of Natural History, Pittsburgh, Pennsylvania, U.S.A.

| | |
|---|---|
| **DMNH** | The Perot Museum of Nature and Science, Dallas, TX, U.S.A. |
| **FMNH** | Field Museum of Natural History, Chicago, Illinois, U.S.A. |
| **IRScNB** | Institute Royal des Sciences Naturelles de Belgique, Brussels, Belgium |
| **IVPP** | Institute of Vertebrate Paleontology and Paleoanthropology, Beijing, China |
| **LACM** | Natural History Museum of Los Angeles County, Los Angeles, California, U.S.A. |
| **MNA** | Museum of Northern Arizona, Flagstaff, Arizona, U.S.A. |
| **NCSM** | North Carolina State Museum of Natural Sciences, Raleigh, North Carolina, U.S.A. |
| **NHMUK** | Natural History Museum of the United Kingdom, London, U.K. |
| **NMC** | Canadian Museum of Nature, Ottawa, Canada |
| **OMNH** | Sam Noble Oklahoma Museum of Natural History, Norman, Oklahoma, U.S.A. |
| **PEFO** | Petrified Forest National Park, Arizona, U.S.A. |
| **PFV** | Petrified Forest Fossil Vertebrate locality |
| **PIMUZ** | Paläontologisches Institut der Universität, Zürich, Switzerland |
| **ROMVP** | Royal Ontario Museum, Ontario, Canada |
| **SAMA** | South Australian Museum, Adelaide, South Australia |
| **SMA** | Sauriermuseum Aathal, Aathal, Switzerland |
| **UCMP** | University of California Museum of Paleontology, Berkeley, California, U.S.A. |
| **UCMZ** | University of Cambridge Museum of Zoology, Cambridge, U.K. |
| **USNM** | National Museum of Natural History, Smithsonian Institution, Department of Vertebrate Zoology, Washington D.C., U.S.A. |
| **USTL** | University of Sciences and Techniques of Languedoc, Montpellier, France |

## ACKNOWLEDGEMENTS

We thank Hunt Consolidated, Inc. for granting accesses to the land that produced this specimen. We thank Dr. Davide Foffa and Erika Goldsmith for discussion and support. We thank Dr. Shuhai Xiao and Prescott Vayda for SEM access and training. We sincerely thank the academic editor, Michela Johnson, and reviewers Stephan Spiekman, Gabriela Sobral, and Jonathan Mitchell for feedback and constructive comments which greatly enriched the manuscript. This is Petrified Forest National Park Contribution no. 98. Views expressed herein are those of the authors and do not represent the views of the United States Government.

### Funding

Funding for this research was provided by the David B. Jones Foundation and the Department of Geosciences at Virginia Tech (Michelle R. Stocker and Sterling J. Nesbitt) and by NSF EAR 1943286 (to Sterling J. Nesbitt). The funders had no role in study design, data collection and analysis, decision to publish, or preparation of the manuscript.

## Grant Disclosures

The following grant information was disclosed by the authors:
David B. Jones Foundation and the Department of Geosciences at Virginia Tech.
NSF EAR: 1943286.

## Competing Interests

The authors declare that they have no competing interests.

## Author Contributions

- Helen E. Burch conceived and designed the experiments, performed the experiments, analyzed the data, prepared figures and/or tables, authored or reviewed drafts of the article, and approved the final draft.
- Hannah-Marie S. Eddins conceived and designed the experiments, performed the experiments, analyzed the data, authored or reviewed drafts of the article, and approved the final draft.
- Michelle R. Stocker conceived and designed the experiments, analyzed the data, authored or reviewed drafts of the article, and approved the final draft.
- Ben T. Kligman analyzed the data, authored or reviewed drafts of the article, and approved the final draft.
- Adam D. Marsh analyzed the data, authored or reviewed drafts of the article, and approved the final draft.
- William G. Parker analyzed the data, authored or reviewed drafts of the article, and approved the final draft.
- Sterling J. Nesbitt conceived and designed the experiments, performed the experiments, analyzed the data, authored or reviewed drafts of the article, and approved the final draft.

## Field Study Permissions

The following information was supplied relating to field study approvals (*i.e.*, approving body and any reference numbers):

Perot Museum of Nature and Science.

## Data Availability

The data is available at Morphosource:

- Media 000607596: Dentary (Partial), 10.17602/M2/M607596
- Media 000623882: Dentary (Partial), 10.17602/M2/M623882.

## New Species Registration

The following information was supplied regarding the registration of a newly described species:

Publication LSID: urn:lsid:zoobank.org:pub:09D15F7E-D5AD-4AC0-BE94-8B7FB5A6DDCF

*Microzemiotes* Genus name LSID: urn:lsid:zoobank.org:act:D13E3E9C-3CFC-41CB-9624-A436826F6BD6

*Microzemiotes sonselaensis* Species name LSID: urn:lsid:zoobank.org:act:0B7BEB75-5F2A-4FD2-B565-CCAE0C53784E.

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
