# Peer review of "A small venomous reptile from the Late Triassic (Norian) of the southwestern United States"

_PeerJ, doi:10.7717/peerj.18279_

## Round 0.1 · original submission · Minor Revisions

Please carefully read through all reviewer comments, which are all minor. Provide more detailed justification(s) on the taxonomic identification as addressed by Reviewers 1 and 3.

·

Basic reporting

Burch et al. describe a new, small-sized dentary of a diapsid reptile. This represents only the third known instance of likely venom-delivering grooves in the dentition of a Triassic reptile, and the only one to also include non-dental remains. As such this represents interesting and valuable new material that merits assignment to a new taxon. The description and interpretation of the material are carried out expertly, and the figures are clear and informative. Although I think the distinction between the new Microzemiotes sonselaensis and Uatchitodon is obvious (i.e. absence of carinae and/or serrations in the teeth of the former), I nevertheless think the authors should clarify this taxonomic distinction more concretely at the end of the “Proposed taxonomy” section in the Discussion. A large size difference can be ontogenetic, and thus does not suffice in distinguishing these forms. The authors also emphasize the diminutive size of Microzemiotes relative to Uatchitodon in an ecological context, calling Microzemiotes “a very small predator”. To provide more context, I think the authors could briefly address the presence or absence of venom-delivering grooves in early ontogenetic stages of extant venomous reptiles, or alternatively, discuss the ontogenetic stage of the holotype of Microzemiotes in more detail. Otherwise, it is unclear whether the authors suggest that the taxon Microzemiotes (and not merely the only known specimen) would have been much smaller than Uatchitodon. Apart from this, I only provide very minor (mostly cosmetic) comments to the manuscript, which can be found in the included annotated PDF.

I hope you find my comments useful in revising your manuscript.

Kind regards,
Stephan Spiekman

Experimental design

The experimental design, including the analysis of the micro CT data, is rigorous. The aim and conclusion of the manuscript are clear.

Validity of the findings

I agree with the main interpretations of the manuscript. As outlined above, the authors could provide a bit more detail for the taxonomic distinction between Microzemiotes and Uatchitodon, as well as for the ontogenetic interpretation of venom-delivering grooves and the holotype of Microzemiotes.

Additional comments

No further comments.

·

Basic reporting

no comment.

Experimental design

no comment

Validity of the findings

no comment

Additional comments

Overall, I think this is a well-written description of a very interesting new specimen. The authors did an excellent job scanning, drawing, and documenting the anatomy of this tiny jaw, and made some excellent and clear figures.

On the main findings of the paper: namely, the contention that this creature is a (1) new (2) venomous (3) diapsid, I think the authors are likely correct, and that they do an excellent job laying out the data supporting each contention. The closest thing to a concern I have is that the size difference between this specimen and Uatchitodon is taken as perhaps too-strong evidence that they're distinct taxa. I agree that they very likely are, but the possibility of a baby Uatchitodon having very small jaws seems to only be raised in a single clause in the last sentence of the manuscript...It may be worth developing that aspect further.

Essentially, I have no major issues with the authors methods or interpretations. I have some minor stylistic recommendations below--I say recommendations, because I think the paper is good as-is, and the authors (and editors) should take the below as I intend them: just recommendations, not major issues.

Lines 64 - 65: This sentence in the abstract is, I think, a bit unclear and focused on the wrong mark. Namely, there are some "bigger" differences (lack of carinae, lack of serrations, shallower rooting) that distinguish this specimen from the known teeth of Uatchitodon better than simply size. A ten-fold difference in size is huge and worth mentioning! But throughout the paper the potential of venom across "multiple body size classes" is sort of asserted vaguely. I don't know that there's enough here to make any meaningful ecological comparisons apt--both this taxon and Uatchitodon are simply too poorly known--but I think the evolutionary distinctiveness is worthwhile and should be the primary emphasis (even though I typically love more ecological emphasis).

Lines 251 - 254: "seem to have homologous arrangements of mandibular anatomy" Not a particularly big issue, but I'm unclear on why this particular feature is ascribed to homology while the intramandibular septum's similarity to Idiosaura is likely convergence (line 273). Note that I don't doubt it, it's just a bit tricky to follow as-written. I wish I could be more helpful, but something about those lines 251-254 threw me a bit.

Lines 263 - 264: You mention the C-shaped septum/edge twice...I think both refer to the same structure, but it's a little unclear

Lines 311 - 318 + Line 335: This is a tricky situation explained, I think, pretty well by the authors. On the one hand, all of the characters used to assign it to Diapsida are plesiomorphies, and the one shared apomorphy it has with a more restricted group (the intramandibular septum) is "dismissed". On the other hand, the reason said apomorphy is dismissed is pretty clear: there are some differences in the exact nature of the character which make it clearly not good evidence for a Sonsela-aged anguimorph fossil. As-written, I think it's good at threading that needle. I do think the way the first bit (lines 311-318) are maybe a bit too formal given it's ultimately a list of plesiomorphies being used as evidence, but I'm not exactly sure what else to recommend in this situation, and again I think the authors did a good job, but if they wanted to tighten the paper any, I think shortening this somewhat would be fine. Few (if any) readers are going to wonder at this assignment, which is appropriately conservative given the ambiguous nature of the evidence.

Lines 382 - 384: Given the extremely large difference here (a pair of lab/ling grooves vs one groove per tooth), I'm skeptical that we can be particularly confident in ascribing a location to the gland here.

Lines 452 - 453: This is the clearest (only?) direct address of the possibility that this specimen could represent a baby Uatchitodon in the paper. Given the lack of carinae, and the lack of carinae. From my reading of lines 230/298 the root-to-crown ratio is something like 0.5, which is shallower than the known roots of Uatchitodon teeth (albeit not by a ton). Some diapsids are known to have pretty profound ontogenetic changes in tooth morphology (see, eg, tegus Tupinambis [formerly]/Salvator Dessem 1985). I find Fig 4 and the writing overall rather convincing that this specimen more likely represents a new taxon rather than a very-small/young Uatchitodon, but I do think the ontogenetic possibilty could be explored more and earlier.

·

Basic reporting

Dear authors,

Congratulations on the manuscript #100155 describing a partial but very characteristic small dentary with associated teeth from the Late Triassic of the United States. The very unusual features of the material justify the erection of a new species and highlight the relevance of the finding. The very diminute size also draws attention to the hidden diversity found among otherwise rarely appreciated fossils.

The manuscript is well written and clear, going straight to the point without being overly speculative, given the limitations of the finding. The figures are also well-made and support the text description quite well.

I have no major concerns about this manuscript, even though I would suggest the authors to better justify their identification of the material as belonging to a diapsid. Although I agree with it, the current justification is based on plesiomorphies that are also found in other reptiles as well as synapsids and even stem amniotes.

Minor comments are listed below.

I hope you find these comments constructive.
Gabriela Sobral

Experimental design

Material and Methods section: Where is the information on the SEM scans?

Validity of the findings

Proposed Taxonomy section: even though I agree with the identification of the material as likely belonging to a diapsid reptile, I feel the justification is very weak. First, I failed to find the referred definition of Diapsida in the works of Ford & Benson in 2019/2020 including the characters listed by the authors. To my knowledge, the clade was not redefined by Ford & Benson and their SD1 indicates these are not unambiguous synapomorphies of the clade in question. Second, these characters were elaborated to potentially identify monophyletic groups among amniotes, and thus the identification of the material is based on plesiomorphic characters, which are phylogenetically uninformative. In fact, this combination of character states is also found among earlier reptiles, some basal synapsids, and even stem-amniotes, so that, based on these characters alone, they cannot be distinguished from other vertebrates of that time and place. The array of plesiomorphic characters is tricky, and the conflicting hypotheses on the position of varanopids is also potentially problematic for the identification of this material, but as it is written, the first paragraph is confusing and wrong. I suggest it be re-written to become clearer and better supported.

Additional comments

Line 89: The cross symbol for extinct taxa is not really necessary, but if the authors find it useful, then please do so throughout manuscript, and not in just a few examples.

Line 127: Do the authors have information on the timing used in the CT scans?

Line 128: The standard is to use the voxel size in µm

Line 177: There is a discrepancy in terminology. Here the authors use “linguo-distally” but in the abstract it is “lingulodistally”. Was this a typo or these are two different spelling for this term? If the latter, please stick to one form only.

Line 178: Lingual wall and labial wall of what? Be very clear to the reader that this refer to a structure of the dentary.

Line 211: Here the authors use “distolingual”. I would suggest to make these terms uniform throughout the manuscript.

Line 212: Lingual wall of what?

Lines 234-235: The listing of Youngina here feels like you are classifying it as an archosauromorph. Please rephrase.

Line 249: Can the authors please justify you think the ventral projection of the dentary is absent instead of missing?

Line 252: Ok, the authors are really identifying Youngina as an archosauromorph. Please correct it.

Line 262: Either lateral and medial or labial and lingual. Please standardise. I would in fact suggest the authors to review the whole text to keep consistency also throughout elements. For instance, labial and lingual are usually reserved for teeth and lateral and medial for bones. Except for labial and lingual walls of the dental shelf, if you are referring to dentary structures, always use medial and lateral, etc.

Line 306: This alternate method of tooth replacement is definitely not universal in all Prolacerta specimens, is it? When analysed with CT scans (Sobral 2023), there appears to have a more complex pattern than the simple alternate method, although some regions of the dentary may conform to this. In fact, this also seems true for stem-diapsids (eg: Hunt 2023). Perhaps this is a thing the authors could add a note on this.

Lines 320-322: This whole sentence feels out of place since you nested varanopids within diapsids in the previous sentence. To highlight the potential problem this diapsid classification brings to the identification of this material, I think the authors should re-write it and make it more explicit.

Lines 322-324: Subthecodont geometry is found among early reptiles and early synapsids, potentially representing the plesiomorphic state for amniotes, and not only diapsids (Bertin et al. 2018).

Line 351: To the reader not familiar with Uatchitodon, it would be interesting to add a few differences between this and the newly described taxon.

Line 367: The new taxon does not have a sharp cutting surface like helopdermatids. The authors compare the new taxon with these lizards quite extensively, but I missed the same level of detail with opitoglyphous snakes, which are only mentioned later on in the discussion, and only very briefly.

Line 375: Here I also missed the mention to opistoglyphous snakes.

Lines 383-385: I found this conclusion speculative and not really adding to the overall discussion.

Line 393: In order TO inflict…

---

## Round 0.2 · accepted · Accept

I commend the authors for having addressed the reviewers' comments in a timely and thorough fashion, and believe that the manuscript is ready for publication.

·

Basic reporting

No further comments

Experimental design

No further comments

Validity of the findings

No further comments

Additional comments

No further comments

·

Basic reporting

This is a much improved version of the manuscript. Several points were corrected or re-written, so that the text and rationale are now much easier to follow and the results more robust. I am very satisfied with the responses given to me by the authors following the first round of reviews and I have nothing more to add to my comments in this new manuscript version.

Experimental design

Nothing to add.

Validity of the findings

Nothing to add.

Additional comments

Nothing to add.